# BMP3 Affects Cortical and Trabecular Long Bone Development in Mice

**DOI:** 10.3390/ijms23020785

**Published:** 2022-01-12

**Authors:** Ivan Banovac, Lovorka Grgurevic, Viktorija Rumenovic, Slobodan Vukicevic, Igor Erjavec

**Affiliations:** 1Department of Anatomy and Clinical Anatomy, School of Medicine, University of Zagreb, 10000 Zagreb, Croatia; ivan.banovac@mef.hr (I.B.); lovorka.grgurevic@mef.hr (L.G.); 2Laboratory for Mineralized Tissues, Center for Translational and Clinical Research, School of Medicine, University of Zagreb, 10000 Zagreb, Croatia; viktorija.rumenovic@mef.hr (V.R.); vukicev@mef.hr (S.V.)

**Keywords:** Bmp3, micro-CT, long bone, cortical bone, growth plate

## Abstract

Bone morphogenetic proteins (BMPs) have a major role in tissue development. BMP3 is synthesized in osteocytes and mature osteoblasts and has an antagonistic effect on other BMPs in bone tissue. The main aim of this study was to fully characterize cortical bone and trabecular bone of long bones in both male and female *Bmp3*^−/−^ mice. To investigate the effect of *Bmp3* from birth to maturity, we compared *Bmp3*^−/−^ mice with wild-type littermates at the following stages of postnatal development: 1 day (P0), 2 weeks (P14), 8 weeks and 16 weeks of age. *Bmp3* deletion was confirmed using X-gal staining in P0 animals. Cartilage and bone tissue were examined in P14 animals using Alcian Blue/Alizarin Red staining. Detailed long bone analysis was performed in 8-week-old and 16-week-old animals using micro-CT. The *Bmp3* reporter signal was localized in bone tissue, hair follicles, and lungs. Bone mineralization at 2 weeks of age was increased in long bones of *Bmp3*^−/−^ mice. *Bmp3* deletion was shown to affect the skeleton until adulthood, where increased cortical and trabecular bone parameters were found in young and adult mice of both sexes, while delayed mineralization of the epiphyseal growth plate was found in adult *Bmp3*^−/−^ mice.

## 1. Introduction

Bone morphogenetic proteins (BMPs) are a large group of growth and differentiation factors that belong to the transforming growth factor-beta (TGF-β) protein superfamily. They have pleiotropic effects in numerous tissues during embryogenesis but also exhibit a prominent role in the regulation of physiological processes and final anatomy of the specimen during postnatal growth and development [1]. All BMP members elicit their cellular effects by ligand-induced activation of type I and II transmembrane serine/threonine kinase receptors. Type II receptors possess constitutional activity and phosphorylate type I receptors that further activate a signaling cascade by phosphorylating receptor-regulated Smads (R-Smads) and mitogen-activated protein kinase (MAPK) [2,3].

BMP3 was first isolated from bone as osteogenin and was determined to be the most abundant BMP in demineralized bone, comprising almost 65% of the total BMP quantity [4,5]. Osteogenin was found to induce osteogenic differentiation both in vitro [6] and in vivo [7] as well as cartilage formation [8]. It was also found in rat mesodermal tissues during embryonic development [9]. Later studies found that osteogenin contained small contaminants of other BMPs and thus expressed osteogenic properties, while pure BMP3 does not promote bone formation [10].

Therefore, the precise function of BMP3 in vivo was further explored in knockout mice (*Bmp3*^−/−^) [11]. *Bmp3*^−/−^ mice are viable and show no obvious skeletal phenotype during development and in the neonatal period. Adult *Bmp3*^−/−^ mice have more bone compared to wild-type mice [10,11], which supports that BMP3 is a negative regulator of bone density [12].

BMP3 is expressed and synthesized in mature osteoblasts and osteocytes [13,14], in the central nervous system of the adult rat [15], the testes of postnatal mice [16], and the bronchial epithelium and collecting tubules of the kidney, intestinal mucosa, perichondrium and periosteum in human development [17]. BMP3 was also shown to be involved in mesenchymal stem cell differentiation, where BMP3/Acvr2b interaction was defined as a key event in BMP mediated osteoblast differentiation. The loss of BMP3 increases colony-forming unit fibroblasts (CFU-F) and osteoblasts (CFU-OB) in bone marrow-derived stem/stromal cell (BMSC) cultures suggesting that BMP3 primarily affects bone formation [14,18,19].

BMP3 is also involved in articular cartilage repair in rabbits [20] and proper regulation of endochondral ossification in the chick limb [21], while overexpression of BMP3 in the developing skeleton alters endochondral bone formation and causes spontaneous rib fractures in mice [22]. Several studies demonstrated the role of BMP3 in regulating bone formation during fracture healing [23,24,25]. Inflammation in rheumatoid arthritis induces BMP3 expression contributing to erosive bone loss [26]. Furthermore, microRNA-450b (miRNA-450b), which was identified as a positive regulator of osteogenic differentiation, was found to directly target and inhibit BMP3 [27]. Interestingly, in vivo loading within a bone chamber model of a rat tibia metaphysis resulted in attenuation of *Bmp3* expression that was followed by differentiation of bone marrow mesenchymal cells into chondrocytes. This finding was the first connection of any growth factor expression under loading conditions [28].

Alterations in BMP3 expression are found in a variety of cancers [29,30,31,32,33], indicating that BMP3 may have other roles besides regulating bone formation.

The aim of this study is to provide a comprehensive characterization of long bones of *Bmp3*^−/−^ mice at various postnatal stages using histology and X-ray microtomography (micro-CT).

## 2. Results

### 2.1. Bmp3 Tissue Localization

Indirect localization of Bmp3 production in P0 mice was visualized using X-gal staining. The blue color from X-gal staining was visible in *Bmp3*^−/−^ mice due to the *LacZ* reporter gene inserted into the *Bmp3* gene yielding a functional β-galactosidase enzyme. X-gal staining was completely absent in WT mice (Figure 1). Bmp3 was observed in lungs, hair follicles, ribs, vertebrae, and skull bones, particularly pronounced in the bones of the viscerocranium and the teeth buds. H&E staining showed no developmental defects in *Bmp3*^−/−^ mice when compared to WT mice.

### 2.2. Effect of Bmp3 on Early Bone Development

The size, shape, and location of skeletal tissue of *Bmp3*^−/−^ mice at the P14 stage appeared normal with no observable defects in development. Nevertheless, qualitative assessment of differential skeletal staining indicated certain differences between WT and *Bmp3*^−/−^ mice in Alizarin Red dye accumulation in the distal femur and proximal tibia as well as in the proximal fibula where observable vesicles were present only in WT mice, indicating that Bmp3 enhances cartilage-to-bone transition and subsequent bone mineralization (Figure 2).

### 2.3. Changes in Cortical and Trabecular Long Bone Parameters of Bmp3^−/−^ Mice Are Not Age- or Sex-Dependent

No significant difference in total body mass was observed between *Bmp3*^−/−^ and WT mice at 8 and 16 weeks of age (Table 1).

The average cortical bone volume fraction (BV/TV) of the femur was up to 40% higher in 8-week-old *Bmp3*^−/−^ mice and up to 22% higher in 16-week-old *Bmp3*^−/−^ mice. The difference was significant in all groups (Figure 3A). The average EV was lower in *Bmp3*^−/−^ mice in all groups, however, the difference was significant only for 16-week-old female mice (Figure 3B). The average Co.Th was higher in *Bmp3*^−/−^ mice and the difference was significant in all groups (Figure 3C). Polar moment of inertia (MMI) was calculated from 3D data and was only significantly higher for 16-week-old *Bmp3*^−/−^ male mice (Appendix A).

Normalized trabecular bone volume revealed that the average trabecular bone volume fraction of the femur was up to 300% higher in 8-week-old *Bmp3*^−/−^ mice and up to 200% higher in 16-week-old *Bmp3*^−/−^ mice. The difference was significant in all groups (Figure 3D). The average Tb.N closely followed the trends observed in trabecular bone volume fraction and was higher in *Bmp3*^−/−^ mice with the difference being significant in all groups (Figure 3E). The average Tb.Th was significantly higher only in 8-week-old *Bmp3*^−/−^, and although it was lower in 16-week-old *Bmp3*^−/−^ mice the difference in this age group was not significant (Figure 3F). Tb. Pf. and SMI were significantly lower only in 16-week-old *Bmp3*^−/−^ female mice (Appendix A).

Representative 3D models highlighting the differences between WT and *Bmp3*^−/−^ mice regarding the cortical and trabecular bone of the femur are shown in Figure 4.

The average cortical bone volume fraction of the tibia was approximately 65% higher in 8-week-old *Bmp3*^−/−^ mice and up to 25% higher in 16-week-old *Bmp3*^−/−^ mice. The difference was significant for all groups, except for 16-week-old female mice (Figure 5A). The average EV was lower in *Bmp3*^−/−^ mice with a significant difference in all groups (Figure 5B). The average Co.Th was higher in *Bmp3*^−/−^ mice and the difference was significant in all groups except for 16-week-old female mice (Figure 5C). The polar moment of inertia (MMI) was significantly higher in *Bmp3*^−/−^ mice for all groups, except 8-week-old female mice (Appendix A).

The average trabecular bone volume fraction of the tibia was up to 250% higher in 8-week-old *Bmp3*^−/−^ mice and up to 200% higher in 16-week-old *Bmp3*^−/−^ mice. The difference was significant in all groups (Figure 5D). The average Tb.N once again closely followed the trends observed in trabecular bone volume fraction and was higher in *Bmp3*^−/−^ mice with a significant difference in all groups (Figure 5E). As observed for the femur, the average Tb.Th was significantly higher in 8-week-old *Bmp3*^−/−^ mice, while in 16-week-old mice no significant difference was observed (Figure 5F). For the tibial trabecular bone, Tb. Pf. was not significantly different between WT and *Bmp3*^−/−^ mice (Appendix A). SMI was significantly lower only in 16-week-old *Bmp3*^−/−^ male mice (Appendix A).

Representative 3D micro-CT reconstructions highlighting the differences between WT and *Bmp3*^−/−^ mice regarding the cortical and trabecular bone of the tibia are shown in Figure 6.

### 2.4. Bmp3 Effect Size and Association Analysis of Long Bone Parameters

The analysis of the effect size using Hedges’ *g* revealed that the increase of BV/TV in *Bmp3*^−/−^ mice was considerably more pronounced in 8-week-old mice than in 16-week-old mice. This indicates that the observed effects of Bmp3 protein deficiency on bone quantity decreased with age mitigating the increase in relative bone volume. The analysis also showed that the increase in BV/TV was slightly more pronounced in female mice (Table 2).

The association between BV/TV and other morphometric parameters was analyzed using correlation and the relationships were modeled using linear regression (Figure 7). The analysis of cortical parameters revealed that there was a high negative correlation (*r* = −0.830) between BV/TV and EV (Figure 7A), and a high positive significant correlation (*r* = 0.863) between BV/TV and Co.Th (Figure 7B), (*p* < 0.001). Linear regression revealed that 68.9% of the variance in BV/TV of cortical bone was predictable from EV (*r*^2^ = 0.689), while 74.5% was predictable from Co.Th (*r*^2^ = 0.745). The analysis of trabecular parameters revealed that there was a very high positive correlation (*r* = 0.986) between BV/TV and Tb.N (Figure 7C), and a moderate correlation (*r* = 0.637) between BV/TV and Tb.Th (Figure 7D), with both correlations being significant (*p* < 0.001). Linear regression revealed that 97.2% of the variance in BV/TV of trabecular bone was predictable from Tb.N (*r*^2^ = 0.972), while 40.6% was predictable from Tb.Th (*r*^2^ = 0.406).

### 2.5. Bmp3 Deficiency Delays Epiphyseal Cartilage Mineralization and Growth Plate Bridging

Histological analysis further confirmed the micro-CT results, where fewer and smaller trabeculae were observed in the distal femur in 16-week-old WT mice compared to *Bmp3*^−/−^ (Figure 8A). Analysis showed that epiphyseal cartilage mineralization and growth plate bridging was delayed in 16-week-old *Bmp3*^−/−^ mice compared to WT littermates. Chondrocyte organization in WT animals was less pronounced with fully defined segments of mineralized cartilage bordering columns of proliferating and hypertrophic chondrocytes. The growth plate in age-matched *Bmp3*^−/−^ mice showed non or low signs of cartilage ossification and growth plate bridging (Figure 8B). Expression of Bmp2, an osteogenic molecule, was higher in the growth plate hypertrophic chondrocytes of *Bmp3*^−/−^ mice (Appendix A). The expression of Runx2, a transcription factor required for chondrocyte maturation and osteoblast differentiation, was also increased in the hypertrophic chondrocytes in *Bmp3*^−/−^ mice (Appendix A).

## 3. Discussion

Previous studies on Bmp3 largely focused on its effect on trabecular bone, particularly that of the vertebrae. In this study, we examined, for the first time, the effect of the *Bmp3* gene on the cortical bone of long bones—the femur and the tibia. Data on the effect of Bmp3 on distal long bones of the lower limb, in particular the tibia, has so far been lacking. In addition, we compared the effect of Bmp3 removal on male and female mice as well as the effect on mice at different stages of postnatal development, providing a more comprehensive overview of the effect of Bmp3 on bone tissue.

The deletion of the *Bmp3* gene was confirmed in P0 mice via the expression of the β-galactosidase enzyme in *Bmp3*^−/−^ mice, in which the *LacZ* gene sequence was inserted in-frame in the first exon of the *Bmp3* gene [14,34]. Previous research demonstrated that *Bmp3* mRNA is expressed in bone, teeth, lungs, kidneys, intestines, and hair follicles [17,35]. Using the *LacZ* gene reporter we confirmed these results on a protein level, as the activity of the β-galactosidase enzyme was detected in bone, hair follicles, and lungs. Bmp3 was visualized by X-gal staining in the flat bones of the skull, the ribs, and the vertebrae. Interestingly, the reporter signal was most pronounced in the viscerocranium, the base of the skull and the teeth buds. It should be noted that the enzyme activity observed in the intestine could potentially originate from externally ingested bacteria (milk) and not from intestinal cells since the mice samples were collected *postpartum*.

*Bmp3*^−/−^ mice had no observable defects and developed at the same rate as WT animals. Bone mineralization was examined 14 days after birth, with differential skeleton staining of the mineralized and cartilaginous portion of the skeleton. The long bones of *Bmp3*^−/−^ mice were more opaque, indicating a higher cartilage replacement rate and more calcium deposition compared to WT mice. This observation, even at a young age, is in accordance with the role of Bmp3 as a negative bone regulator [12]. The lack of a major effect of *Bmp3* gene removal on mouse size indicates the existence of redundancy for BMP inhibition during bone growth and development [36,37]. Nevertheless, Bmp3 plays a major role in BMP inhibition, mostly due to its abundance [38].

Although muscle tissue was not the focus of this research, the fact that no significant difference in body mass was observed between *Bmp3*^−/−^ and WT mice warrants further investigation of the possible effect of Bmp3 on skeletal muscles [39]. Even though excess bone tissue could require more muscle mass for movement, we did not observe any significant changes in body mass in *Bmp3*^−/−^ mice, and thus, indirectly no significant increase in muscle mass. This finding could possibly be explained by the predominantly sedentary life of laboratory mice which might cause any existing effects on muscle mass to be less pronounced.

Experimental research has shown that Bmp3 is expressed in the hypertrophic cell layer of the femur growth plate, which supports findings that Bmp3 is involved in trabecular bone growth [24]. Furthermore, significant effects of genetic manipulation on animal phenotype at a young age were observed in numerous mouse models [40]. The findings in our study suggest that the differences in bone mineralization in *Bmp3*^−/−^ mice observed at P14 may persist until maturity. At P14 we analyzed only male mice due to the fact that sexual dimorphism in bone length in mice is not observed prior to three weeks of age [40].

Nevertheless, it might be reasonable to assume that functional knockout animals could display differences in bone metabolism between sexes at a mature age due to different responses to sex hormones [41]. For example, deletion of estrogen receptors reveals a regulatory role for estrogen receptors-beta in bone remodeling in females, but not in males. In addition, Bmp3 could affect male and female mice differently due to sexual dimorphism in mice. In this study, we examined the potential sex differences in Bmp3 bone metabolism regulation by comparing the effects of Bmp3 deletion in both male and female mice. When the effect of mice size on bone volume was eliminated by normalizing it according to tissue volume, a significant increase in bone volume fraction (BV/TV) in *Bmp3*^−/−^ mice was observable in both sexes and in both analyzed age groups. This supports the notion that the fundamental inhibitory effect of Bmp3 is similar in both sexes, even though effect size analysis suggests that it might be slightly more pronounced in female mice.

Since our findings demonstrated a significant effect of Bmp3 on cortical bone volume, it is important to note that the formation of cortical bone comprises of two distinct processes: diaphyseal cortical bone is formed by sub-periosteal apposition, while metaphyseal cortical bone is formed by trabecular coalescence [42] and is more pronounced in male mice [43]. Bone corticalization requires local SOCS3 activity and is promoted by androgen action via interleukin-6. In our research, the effect of *Bmp3* deletion on cortical bone parameters followed the same trends in both male and female mice. In general, physiological decrease in femoral cortical thickness occurs in both male and female mice as a result of decreased periosteal formation, increased endosteal resorption, medullary expansion, increased osteocyte DNA damage, cellular senescence, and increased levels of RANKL [44]. Higher responsiveness of the endocortical bone surface to mechanical loading could also serve as a mechanism for increased bone resorption due to lower animal activity in older age [45]. In our study, the cortical bone analysis revealed that both male and female *Bmp3*^−/−^ mice had, on average, a lower endosteal volume which corresponds to a smaller medullar canal, although the difference was significant only in female mice (16-week-old mice for the femur and both age groups for the tibia). Other studies found that the porosity of the femoral cortical bone in mice increased with age and had a significantly higher incidence in females, which, along with increased endosteal resorption, led to a reduction in cortical thickness in older animals [46]. In addition, *Bmp3*^−/−^ mice in our study had a higher femoral diaphyseal cortical thickness with significant differences for most groups, with the exception of the tibia in 16-week-old mice. The fact that *Bmp3*^−/−^ mice had a higher cortical thickness, but a lower endosteal volume suggests that endosteal resorption in *Bmp3*^−/−^ mice was decreased compared to WT animals.

In addition to the effect on cortical bone, our study also confirmed the effect of the removal of the *Bmp3* gene on the trabecular bone of long bones in both male and female mice. Trabecular bone analysis of the femur revealed that *Bmp3*^−/−^ mice had an approximately two times higher bone volume fraction than WT mice, which is in line with results from previous studies [11,12]. A similar trend was observed for the trabecular bone of the tibia, further establishing the effect of the *Bmp3* gene on long bones. From the trabecular parameters that reflect BV/TV, only the trabecular number was shown to have a consistent significant increase in *Bmp3*^−/−^ mice. Trabecular thickness increased significantly only in 8-week-old *Bmp3*^−/−^ mice, while it decreased in 16-week-old *Bmp3*^−/−^ mice, albeit not significantly. Furthermore, the correlation coefficient between bone volume fraction and trabecular number is much higher than the correlation coefficient between bone volume fraction and trabecular thickness. This indicates that the increase in trabecular bone volume in *Bmp3*^−/−^ mice is primarily due to an increase in the number of trabeculae, while the changes in trabecular thickness contribute much less to the increase in trabecular bone volume in *Bmp3*^−/−^ mice.

Histological analysis of the distal femur of 16-week-old mice revealed that more trabeculae were present in *Bmp3*^−/−^ mice, which was in line with the quantitative findings from the micro-CT analysis. Furthermore, no mineralized cartilage was observed in the epiphyseal growth plate of *Bmp3*^−/−^ mice. This finding is relevant because, unlike in humans, the epiphyseal growth plate in mice does not close during puberty [46], but rather undergoes extensive cartilage mineralization causing plate bridging and subsequent thinning of the growth plate [47,48]. We found signs of epiphyseal growth plate bridging only in WT mice, which indicates that the process is either completely absent in *Bmp3*^−/−^ mice, or at least significantly delayed not to be observable at 16 weeks of age. It is interesting to note that previous research found *Bmp3* mRNA expression in hypertrophic chondrocytes, but not in other parts of the growth plate [24]. Altogether, this supports the notion that Bmp3 expression is attenuated prior to chondrocyte differentiation from bone marrow osteoprogenitor cells [28]. Since there is no difference in the length of long bones between *Bmp3*^−/−^ and WT mice, it can be concluded that Bmp3 deficiency does not significantly affect long bone cortical growth, thus, it likely affects neither chondrocyte differentiation nor proliferation. Instead, Bmp3 deficiency appears to primarily affect chondrocyte hypertrophy and extracellular matrix deposition in the epiphyseal region, resulting in increased formation of trabecular bone, with higher expression of Runx2 in that region. This is caused by an excess availability of osteoinductive molecules, such as BMP2 in *Bmp3*^−/−^ mice.

The aforementioned results demonstrated that Bmp3 in mice has a significant role in the regulation of long bone formation, affecting both cortical and trabecular bone respectively. Similar effects of *Bmp3* on long bones were observed in both male and female mice, excluding a significant role of sex hormones in the mechanism of its action. These results further corroborate a significant regulatory role of Bmp3 deficiency in bone metabolism leading to an increased trabecular and cortical bone volume of long bones in mice.

## 4. Materials and Methods

### 4.1. Animal Model

The study was carried out on mice of the C57BL/6NTac strain, produced by Regeneron Pharmaceuticals (Tarrytown, NY, USA) using VelociGene technology [14,34]. The mouse model was generated by the in-frame insertion of the β-galactosidase (*LacZ*) reporter gene in the first exon of the *Bmp3* gene. The breeding and housing were carried out in the Laboratory for Mineralized Tissues, School of Medicine, University of Zagreb, Croatia. They were housed under controlled temperature (23 ± 2 °C), humidity (55 ± 10%), and light cycle (12 h light/12 h dark) with food and water ad libitum.

The study was approved by the institutional (School of Medicine, University of Zagreb) Ethics Committee under three different projects (380-59-10106-13-195/168,380-59-10106-21-111/52 and IP-2020-02-5960) and national Ethics Committee (EP327/2021) and was conducted according to the national acts and guidelines for care and use of laboratory animals.

### 4.2. Genotype Determination

To determine the genotype of each mouse, tissue samples were collected from the tail tips at the age of 3 weeks. They were then incubated in a lysis buffer containing proteinase K at 56 °C overnight. After incubation, the DNA was isolated using standard protocol [49]. The concentration and purity of the DNA were examined using a Biophotometer (Eppendorf, Hamburg, Germany). The primers used for genotyping were: *Bmp3* (F-GAAGTAGAGCGGTGCGACAGCA, R-AAGGTCCCTACAGTGTACCGCCA; 496 bp) and *LacZ* (F-TTTCCATGTTGCCACTCGC, R-ACCGCACGATAGAGATTCGG; 264 bp). PCR was conducted using the EmeraldAmp^®^ MAX PCR set (Takara, Shiga, Japan) in Thermal Cycler 2400 (Perkin Elmer, Waltham, MA, USA). The multiplied DNA sequences were separated using 1.5% agarose gel electrophoresis, while 0.01% ethidium bromide (Millipore-Sigma, Munich, Germany) was added to visualize the DNA bands. The gel was imaged in a UV transilluminator (Appligene Oncor, Illkirch, France) with a wavelength of 300 nm.

### 4.3. Bmp3 Tissue Expression

To examine the tissue localization of the Bmp3 protein, the β-galactosidase (LacZ) reporter protein was used. Mice at P0 (bred from WT or *Bmp3*^−/−^ parents exclusively) were terminated using deep ether anesthesia and tissue samples were fixed in 10% paraformaldehyde. Samples were then cryoprotected for three days in phosphate-buffered saline solutions of increasing sucrose concentrations (10%, 20%, and 30%), after which they were embedded in OCT Tissue Tek (Sakura, Osaka, Japan) and frozen at −80 °C. Frozen samples were then transferred to −20 °C and prepared for cutting on a CM1850 cryotome (Leica, Wetzlar, Germany) at a thickness of 20 µm. Cross-sections were stained using a standard procedure for X-gal [50] or standard H&E staining. Samples were imaged using a BX-53 microscope (Olympus, Tokyo, Japan).

### 4.4. Differential Skeleton Staining

Cartilage and skeleton staining were adopted from a previous protocol [51]. Mice at P14, (bred from WT or *Bmp3*^−/−^ parents exclusively) were terminated using deep ether anesthesia. A small amount of skin was taken for final genotype confirmation. Prior to staining, they were placed in hot water to facilitate skin removal, after which they were eviscerated, and their eyes were removed. Samples were fixed by placing them in Formalin-Acetic-Alcohol (FAA) for 45 min. Cartilage was stained by Alcian Blue, after which bone tissue was stained with Alizarin Red. Samples were then destained in 0.5% KOH and glycerin and imaged using an SZX10 stereo microscope (Olympus, Tokyo, Japan).

### 4.5. Bone Characterization

The experimental group consisted of a total of 20 *Bmp3* knockout mice (*Bmp3*^−/−^) divided by age and sex into the following groups: 8-week-old males (*n* = 4), 8-week-old females (*n* = 4), 16-week-old males (*n* = 6) and 16-week-old females (*n* = 6). Age- and sex-matched wild-type (WT) mice of the same strain were used as controls. The femur and tibia of the left hind leg were collected, the surrounding soft tissue was removed, and the bones were fixed in 4% formaldehyde for 48 h [52].

### 4.6. Micro-CT Analysis

The extracted left femur and tibia were imaged using a micro-CT scanner (SkyScan 1076, Bruker, Kontich, Belgium) at a 9 µm isotropic voxel size with a voltage of 50 kV and an electric current of 200 µA. Beam-hardening was reduced using a 0.5 mm thick aluminium filter. The rotational shift was set to 0.5° in an area of 198° for every sample. After acquisition, image reconstruction was performed using the NRecon software (Bruker, Kontich, Belgium).

The acquired sample data sets were analyzed using the CTAn software (Bruker, Kontich, Belgium). Morphometric analysis for trabecular bone analysis was performed on the distal femur and proximal tibia, while the midshafts of the femur and tibia were used for cortical bone analysis [52,53]. Morphometric analysis of the femur was conducted using the distal growth plate as a reference point. To avoid primary spongiosa, the offset from the growth plate was 50 slices for trabecular bone, while a 350 slices offset was used for cortical bone. The volume of interest was 50 slices for cortical bone and 100 slices for trabecular bone. Morphometric analysis of the tibia was performed in the same manner, with the exception of using the proximal growth plate as a reference point, with the same parameters as for the femur. The following parameters were analyzed for the cortical bone: bone volume fraction (BV/TV), endosteal volume (EV), cortical thickness (Co.Th), and polar moment of inertia (MMI). EV was calculated by subtracting BV from TV and it represents the volume inside the medullary cavity [54]. The following parameters were analyzed for the trabecular bone: BV/TV, trabecular number (Tb.N), trabecular thickness (Tb.Th), trabecular pattern factor (Tb.Pf), and structure model index (SMI).

### 4.7. Histological and Immunohistochemical Analysis

Following hind limb fixation and micro-CT scanning, femur samples were decalcified using 14% EDTA in 4% formalin solution for 28 days. The solution was replaced every 2 days. Decalcified specimens were embedded in paraffin and cut longitudinally into 6 μm thick sections using a microtome (Leica, Wetzlar, Germany). Sections were processed using a standard protocol for hematoxylin and eosin (H&E) staining [55]. Images of H&E-stained sections were acquired using an Olympus BX53 light microscope (Olympus, Tokyo, Japan) and image analysis was performed in ImageJ (NIH, Bethesda, MD, USA). The total growth plate (GP) height of the distal femur was measured at 10 different points along the length of the GP based on established cell morphology in similar locations on eight individual animals (four WT and four *Bmp3*^−/−^) [56]. The number of bridging elements of the growth plate was counted, and the width of each bridging mineralized tissue was measured. Immunohistochemical analysis was performed using a standard protocol. In brief, heat-induced epitope retrieval (HIER) was performed in a sodium citrate buffer at pH 6,0. A mouse- and rabbit-specific HRP/DAB IHC Detection Kit—Micro-polymer (ab236466, Abcam, Waltham, MA, USA) was used in all procedures. Sections were pretreated with Hydrogen Peroxide Block and Protein Block reagents from the kit to eliminate endogenous peroxidase activity and nonspecific protein binding. Bmp2 (ab14933, Abcam, Waltham, MA, USA) and Runx2 (ab192256, Abcam, Waltham, MA, USA) antibodies were used to assess protein expression in bone tissue. Slides were counterstained with hematoxylin and mounted using ImmunoHistoMount™ (Sigma-Aldrich, Burlington, MA, USA). Images were obtained using an Olympus BX53 light microscope (Olympus, Tokyo, Japan).

### 4.8. Data Management

The data were analyzed using GraphPad Prism ver. 8.3.0 and represented as arithmetic mean ± standard deviation. For comparison of *Bmp3*^−/−^ and WT groups, the unpaired two-tailed Welsh’s *t*-test with presumed unequal variances was used and *p* < 0.05 was considered statistically significant [57]. Mann–Whitney’s test was used to compare the differences in the number of bridging elements between *Bmp3*^−/−^ and WT mice. To estimate the effect size of the differences, the measure Hedges’ *g* was calculated for the parameter BV/TV. Hedges’ *g* provides a standardized mean difference and enables the evaluation of effect sizes between different comparisons.

A correlation was used to determine the association between morphometric parameters, with Pearson’s correlation coefficient (*r*) used as a measure for the strength of the association. The *t*-test was used to determine whether the correlation coefficient was significantly different from zero with *p* < 0.05 considered to be statistically significant [58]. Linear regression was used to determine the proportion of the variance in BV/TV that is predictable from the variance of another morphometric parameter, which is expressed as the coefficient of determination *r*^2^.

## 5. Conclusions

Bmp3 gene deletion in mice caused an increase in the cortical and trabecular bone volume of the distal femur and proximal tibia, further confirming the inhibitory role of Bmp3 in bone tissue. An increase in bone calcification was observed in P14 *Bmp3*^−/−^ mice, while an increase in bone volume was observed in 8-weeks- and 16-weeks-old *Bmp3*^−/−^ mice in both sexes. The increase in cortical bone volume and cortical thickness in *Bmp3*^−/−^ mice was mostly a consequence of decreased endosteal resorption, while periosteal apposition seemingly remained unaffected. Trabecular bone was increased due to the impaired inhibitory effect of Bmp3 on cartilage to bone transition in the epiphyseal growth plate. Comprehensive analysis of cortical and trabecular bone suggests that Bmp3 antagonizes the effects of osteogenic BMPs in bone, preventing uncontrolled bone formation and offering a mechanical link to growth factor expression in the adaptation of bone to load.

## Figures and Tables

**Figure 1 ijms-23-00785-f001:**
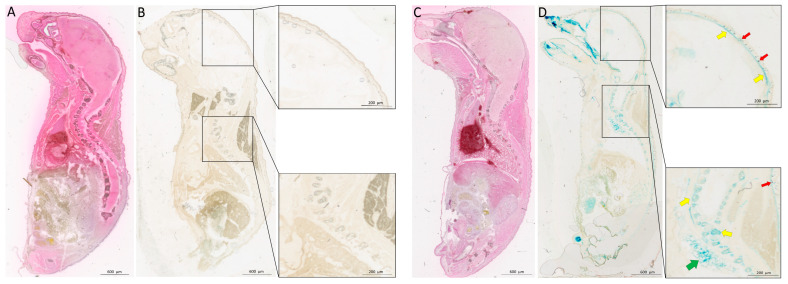
Expression of β-galactosidase (LacZ) in P0 mice. H&E staining shows no abnormalities in *Bmp3*^−/−^ mice (**C**) compared to WT (**A**) mice. X-gal staining shows no LacZ reporter protein in WT (**B**), while in *Bmp3*^−/−^ mice (**D**) localization of LacZ reporter protein can be observed in bone (yellow arrows), hair follicles (red arrows), and lungs (green arrow). X-gal staining was particularly pronounced in the viscerocranium and the base of the skull.

**Figure 2 ijms-23-00785-f002:**
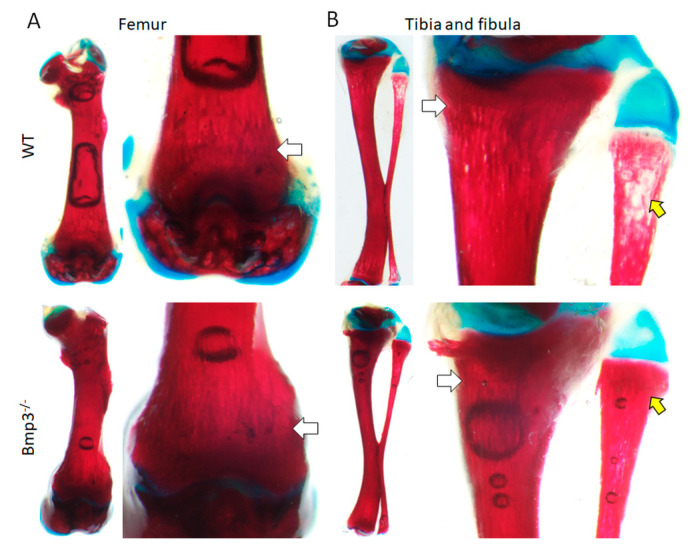
Differential skeletal staining of long bones in P14 mice. Mineral deposition (white arrows) in the distal femur (**A**) and proximal tibia (**B**) appears more pronounced in *Bmp3*^−/−^ mice compared to WT mice. The effect is the most pronounced in the proximal fibula (yellow arrow) where observable vesicles were present in all WT mice, but not in any of the analyzed *Bmp3*^−/−^ mice.

**Figure 3 ijms-23-00785-f003:**
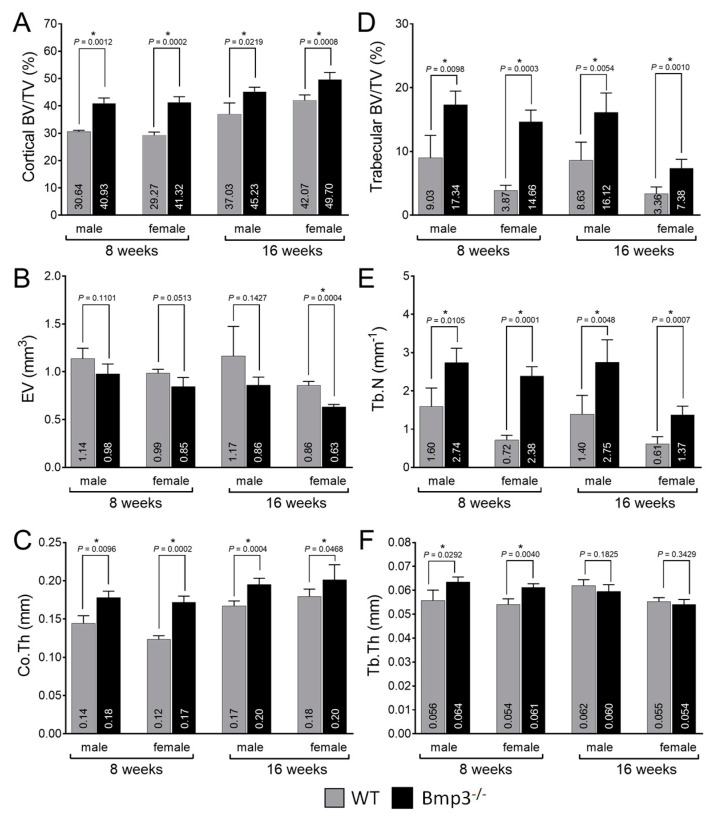
Morphometric analysis of the distal femur of *Bmp3*^−/−^ and WT mice at 8 and 16 weeks of age. (**A**) BV/T, (**B**) EV and (**C**) Co.Th for femoral cortical bone. (**D**) BV/TV, (**E**) Th.N and (**F**) Tb.Th for femoral trabecular bone. Statistically significant differences are indicated by *.

**Figure 4 ijms-23-00785-f004:**
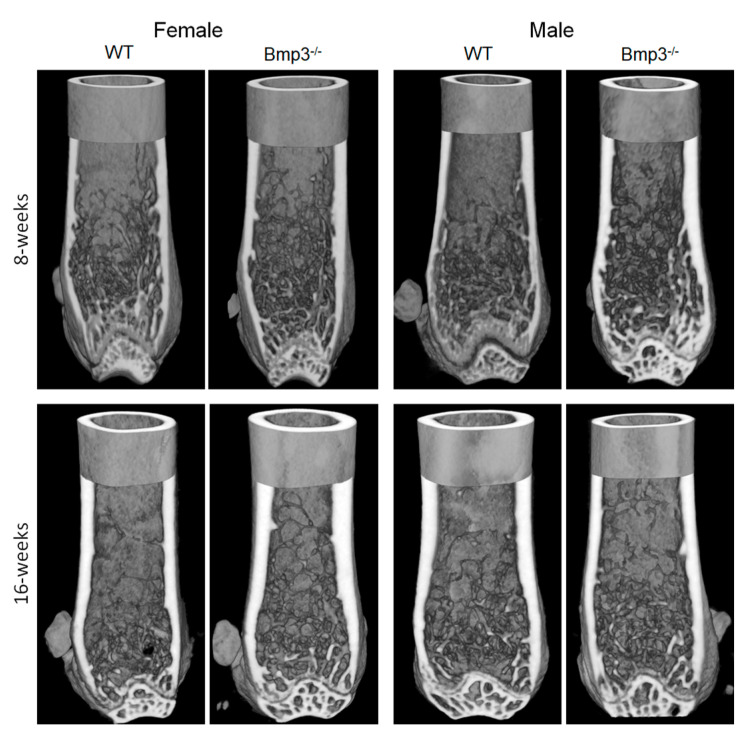
Femur cortical and trabecular bone render of 8-week- and 16-week-old female and male *Bmp3*^−/−^ and WT mice.

**Figure 5 ijms-23-00785-f005:**
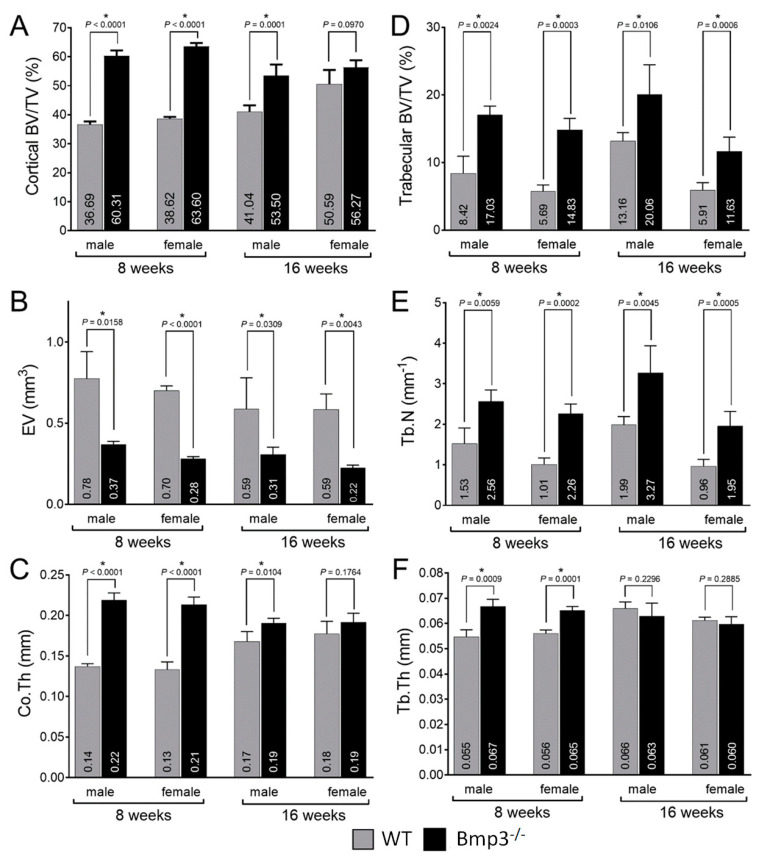
Morphometric analysis of the proximal tibia of *Bmp3*^−/−^ and WT mice at 8 and 16 weeks of age. (**A**) BV/TV, (**B**) EV and (**C**) Co.Th for tibial cortical bone. (**D**) BV/TV, (**E**) Th.N and (**F**) Tb.Th for tibial trabecular bone. Statistically significant differences are indicated by *.

**Figure 6 ijms-23-00785-f006:**
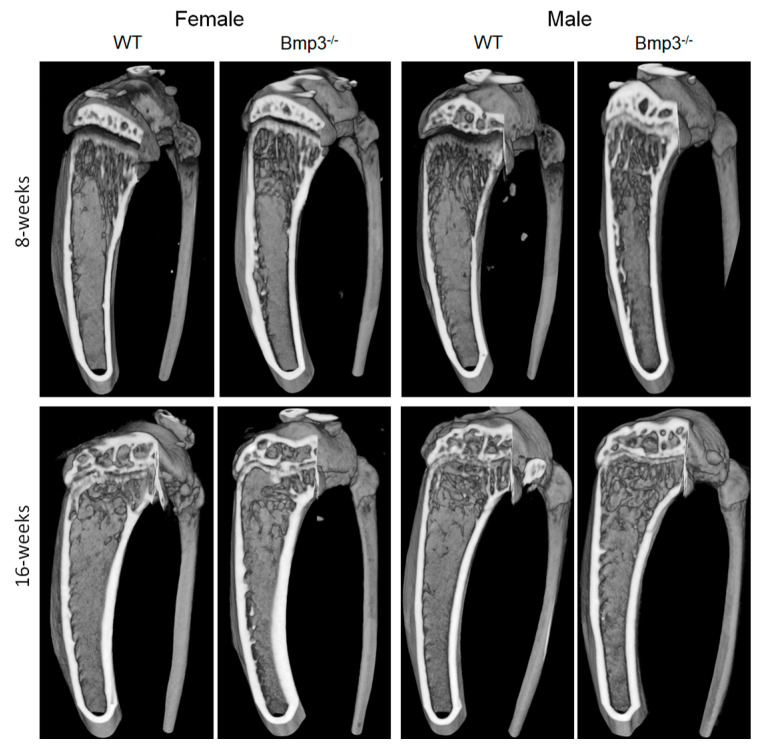
Tibia cortical and trabecular bone render of 8-week- and 16-week-old female and male *Bmp3*^−/−^ and WT mice.

**Figure 7 ijms-23-00785-f007:**
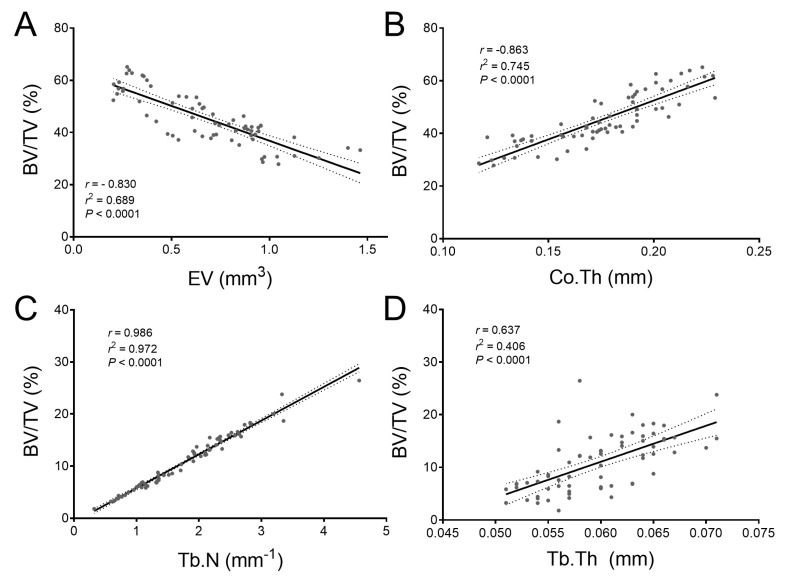
Scatter plots showing correlation between BV/TV and other morphometric parameters. (**A**) EV. (**B**) Co.Th. (**C**) Tb.N. (**D**) Tb.Th. The full line represents the line of best fit derived from simple linear regression, while the dotted lines represent the 95% confidence intervals.

**Figure 8 ijms-23-00785-f008:**
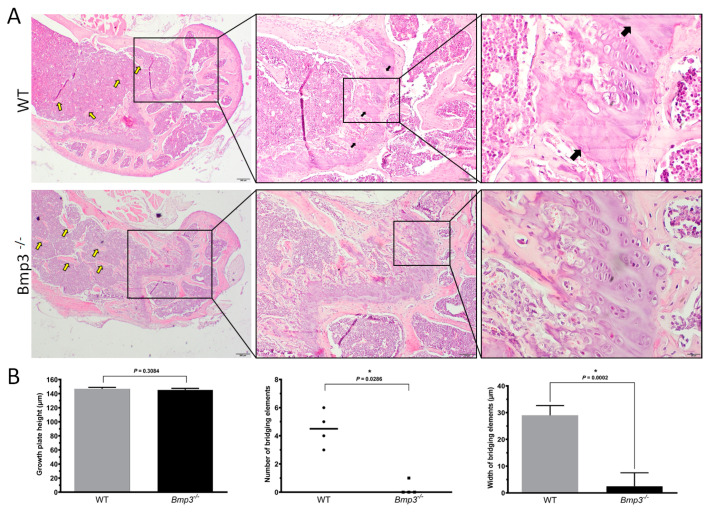
Histology of the mouse femur at 16 weeks of age. The difference in trabecular bone between WT and *Bmp3*^−/−^ mice (yellow arrows) is in concordance with the results of micro-CT analysis. Bridging regions of mineralized cartilage in the epiphyseal growth plate were observed in WT mice (black arrows), but not in *Bmp3*^−/−^ mice (**A**). No difference in growth plate height was observed between *Bmp3*^−/−^ and WT mice. Bridging element regions were significantly decreased in *Bmp3*^−/−^ mice (**B**). Statistically significant differences are indicated by *.

**Table 1 ijms-23-00785-t001:** Total body mass (in g) of young and adult *Bmp3*^−/−^ and WT mice in males (M) and females (F). Data are represented as the mean ± standard deviation.

	8 Weeks		16 Weeks	
	WT	*Bmp3* ^−/−^	WT	*Bmp3* ^−/−^
**M**	20.68 ± 1.49	19.30 ± 2.67	25.73 ± 0.49	23.98 ± 1.91
**F**	18.62 ± 1.22	18.31 ± 2.67	20.31 ± 1.14	21.65 ± 3.15

**Table 2 ijms-23-00785-t002:** Values of Hedges’ *g*, comparing *Bmp3*^−/−^ and WT mice, for normalized bone volume (BV/TV).

		Femur		Tibia	
		Cortical Bone	Trabecular Bone	Cortical Bone	Trabecular Bone
**8 weeks**	**M**	6.6931	2.8748	15.7512	4.2907
	**F**	7.1881	7.6275	26.5651	6.6429
**16 weeks**	**M**	2.8073	2.9811	4.065	2.0948
	**F**	3.2661	3.1393	1.5804	3.1762

## Data Availability

Non applicable.

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
