# Peer review of "BMP3 Affects Cortical and Trabecular Long Bone Development in Mice"

_ijms, 2022, doi:10.3390/ijms23020785_

Round 1

Reviewer 1 Report

The work of Banovac et al. explored the antagonistic role that BMP3 plays in bone development and its homeostasis. Through a comparative study between BMP3 knock out and WT C57BL/6NTac mice, the authors conclude that lack of BMP3 expression increased cortical and trabecular bone while delayed mineralization of the epiphyseal growth plate.

Reviewer suggestions:

M&M:

Authors should use “C57BL/6NTac” instead of “C57BL/6Ntac”.

Figure 1:

The author should adequately justify why olfactory and nasopharyngeal region were positive for X-gal staining in BMP3 -/- mice. The staining of those regions was more intense than that found in the bone.

Figure 2:

Differences between WT and BMP3 -/- mice were observed in Alizarin Red staining of proximal fibula along with distal femur and proximal tibia. The authors showed representative images from selected individuals. To gain reliability, quantified data should be presented (for example, by image segmentation methods) from a number of individuals that reaches statistical significance.

Figure 8:

Again, the authors showed representative images of two individuals in a histological comparison. This type of representation can be a source of bias. For the aforementioned observations to gain validity, it would be necessary to proceed as indicated for Figure 2.

Did the author check any bone formation and resorption markers in serum of studied mice? This type of study could provide an important dimension to understand the implications of BMP3 deficiency in the homeostasis of the long bone.

Did the author study BMP2 or OCN expression in the long bone of BMP3 -/- (for example by immunohistochemical methods)? This type of study would serve to validate that the effects observed in the long bone of BMP3 -/- mice are mainly caused by the excessive bioavailability of gene products that stimulate bone mineralization.

Have the authors characterized the functional profile, at a mechanical level, of long bones obtained from BMP3 -/- mice? This type of study is essential to understand the biological fitness implications of this type of over-mineralized bone.

What about the intramembranous ossification? What implications, if any, do the conclusions of the presented study have on this bone formation process?

Author Response

Authors should use “C57BL/6NTac” instead of “C57BL/6Ntac”.

RESPONSE: We thank the reviewer for the comment. We have corrected the name of the breed in the text.

Figure 1:

The author should adequately justify why olfactory and nasopharyngeal region were positive for X-gal staining in Bmp3-/- mice. The staining of those regions was more intense than that found in the bone.

RESPONSE: We thank the reviewer for the comment. We have described the more intense coloration of the skull bones and teeth buds in the revised version of the manuscript. Although, in this research, our focus was on long bones, more expression of BMP3 in skull bones is interesting for future research.

Figure 2:

Differences between WT and BMP3 -/- mice were observed in Alizarin Red staining of proximal fibula along with distal femur and proximal tibia. The authors showed representative images from selected individuals. To gain reliability, quantified data should be presented (for example, by image segmentation methods) from a number of individuals that reaches statistical significance.

RESPONSE: We thank the reviewer for this valuable insight. Alizarin red/Alcian blue staining is used for delineation of cartilage from bone and we have not found any method, in the current literature, that describes segmentation and quantification of Alizarin red stained bone. We agree with the reviewer that statements regarding the amount of mineralization cannot be derived by qualitative analysis alone and have accordingly revised the text to state what can be clearly derived from qualitative assessment of Alizarin red staining. The key difference between Bmp3-/- and WT mice was observed in the proximal fibula, where vesicles with no mineral deposition were found in all analyzed WT mice at P14. In addition, such vesicles were absent in all analyzed Bmp3-/- mice at P14.

Figure 8:

Again, the authors showed representative images of two individuals in a histological comparison. This type of representation can be a source of bias. For the aforementioned observations to gain validity, it would be necessary to proceed as indicated for Figure 2.

RESPONSE: We thank the reviewer for the comment. In the Figure 8. we have added graphs with quantification of the growth plate height and we have assessed the bridging mineralized tissue (elements) in the growth plate by counting them and measuring their width.

Did the author check any bone formation and resorption markers in serum of studied mice? This type of study could provide an important dimension to understand the implications of BMP3 deficiency in the homeostasis of the long bone.

RESPONSE: We thank the reviewer for addressing issue. We are aware that measuring bone remodeling serum markers would give us additional insight in the effect BMP3 exerts on bone, however, we have not included the bone serum markers in this experiment.

Did the author study BMP2 or OCN expression in the long bone of BMP3 -/- (for example by immunohistochemical methods)? This type of study would serve to validate that the effects observed in the long bone of BMP3 -/- mice are mainly caused by the excessive bioavailability of gene products that stimulate bone mineralization.

RESPONSE: We thank the reviewer for the comment. We have now conducted immunohistochemical analyses of Bmp2 and Runx2 proteins in bone tissue of Bmp3-/- and WT mice, and added the results in the revised manuscript. Runx2 is highly specific for hypertrophic chondrocytes and osteoblasts, so we decided to use it instead of OCN.

Have the authors characterized the functional profile, at a mechanical level, of long bones obtained from BMP3 -/- mice? This type of study is essential to understand the biological fitness implications of this type of over-mineralized bone.

RESPONSE: We thank the reviewer for the question. We have not conducted biomechanical testing of bones, but  to supplement this information we added the data for indirect biomechanical parameters from micro-CT analyses which include structure model index (SMI) and trabecular pattern factor (Tb.Pf) for trabecular bone, and polar moment of inertia (MMI) for cortical bone. Our prior studies have shown that similar analyses were comparable to direct mechanical testing in rodents. (Sampath TK et al. Sevelamer Restores Bone Volume and Improves Bone Microarchitecture and Strength in Aged Ovariectomized Rats; Endocrinology; 2008.).

What about the intramembranous ossification? What implications, if any, do the conclusions of the presented study have on this bone formation process?

RESPONSE: We thank the reviewer for this important question. Since our focus in this research was on long bones, we did not assess intramembranous ossification and, therefore, cannot adequately extrapolate our results on this ossification process. However, taking into account the extent of Bmp3 expression in skull bones, we plan to investigate the intramembranous ossification in flat bones of the skull and in the clavicula as well in fracture healing in future experiments.

Reviewer 2 Report

The study appears to be carefully done by respected research Erdelyi in the 

However, the results have potential implications which haven’t been explored and should be addressed in this paper.

  1. Are there effects of BMP3 deletion on bone biomechanical properties ?  If so, there may be implications for fracture risk as well as for fracture repair.
  2. You have had a golden opportunity to study the effects of excess bone on muscle mass and function. 
    While I understand that this was not the primary focus of your work, it should at least be addressed in the Discussion as a future direction .

Author Response

The study appears to be carefully done by respected research Erdelyi in the 

However, the results have potential implications which haven’t been explored and should be addressed in this paper.

  1. Are there effects of BMP3 deletion on bone biomechanical properties ?  If so, there may be implications for fracture risk as well as for fracture repair.

RESPONSE: We thank the reviewer for the question. We have not conducted biomechanical testing of bones, but  to supplement this information added the data for indirect biomechanical parameters from micro-CT analyses which include structure model index (SMI) and trabecular pattern factor (Tb.Pf) for trabecular bone, and polar moment of inertia (MMI) for cortical bone. Our prior studies have shown that similar analyses were comparable to direct mechanical testing in rodents. (Sampath TK et al. Sevelamer Restores Bone Volume and Improves Bone Microarchitecture and Strength in Aged Ovariectomized Rats; Endocrinology; 2008.). We do plan including functional biomechanical testing and computer modeled testing by finite element analyses (FEA) in future experiments.

  1. You have had a golden opportunity to study the effects of excess bone on muscle mass and function. 
    While I understand that this was not the primary focus of your work, it should at least be addressed in the Discussion as a future direction.

RESPONSE: We thank the reviewer for the comment. Although the link between bone and muscle tissue is well established in regard to locomotor functions, the removal of Bmp3 did not result in increased total mass, therefore we did not focus on assessing the direct relationship between bone and muscle mass in this research. We have addressed the possible implications of excess bone on muscle mass in the Discussion.

Round 2

Reviewer 1 Report

No additional comments to those already presented in the first review.